# The Effects of Housing on Growth, Immune Function and Antioxidant Status of Young Female Lambs in Cold Conditions

**DOI:** 10.3390/ani14030518

**Published:** 2024-02-05

**Authors:** Jin Xiao, Wenliang Guo, Zhipeng Han, Yuanqing Xu, Yuanyuan Xing, Clive J. C. Phillips, Binlin Shi

**Affiliations:** 1College of Animal Science, Inner Mongolia Agricultural University, Hohhot 010018, China; yaojinxiao@aliyun.com (J.X.); 18686197338@163.com (W.G.); hzp4639@163.com (Z.H.); happyxyq@yeah.net (Y.X.); xingyuanyuan2014@163.com (Y.X.); 2Curtin University Sustainability Policy (CUSP) Institute, Curtin University, Perth, WA 6845, Australia; clive.phillips@curtin.edu.au; 3Institute of Veterinary Medicine and Animal Science, Estonia University of Life Sciences, Kreutzwaldi 1, 51014 Tartu, Estonia

**Keywords:** low temperature, housing condition, heat shock protein, immune function, young female lambs, antioxidant status, gene expression, sheep

## Abstract

**Simple Summary:**

In the winter in northern China, the type of housing or outdoor management may affect the growth and health of sheep, especially if they are young. We compared growth performance, immune function and antioxidant status of female lambs housed there in an enclosed building, a polytunnel or kept outdoors for 28 days. Compared to lambs kept outdoors at −14 °C, those in the enclosed house and polytunnel had improved growth rates, immune function and antioxidant status. We conclude that the polytunnel is best for the sheep’s welfare, followed by the enclosed house, with the outdoor treatment being the worst.

**Abstract:**

Cold conditions in northern China during winter may reduce sheep growth and affect their health, especially if they are young, unless housing is provided. We allocated 45 two-month-old female lambs to be housed in an enclosed building, a polytunnel, or kept outdoors, for 28 days. The daily weight gain and scalp and ear skin temperature of outdoor lambs were less than those of lambs that were housed in either a house or polytunnel; however, rectal temperature was unaffected by treatment. There was a progressive change in blood composition over time, and by the end of the experiment, outdoor lambs had reduced total antioxidant capacity (T-AOC), catalase (CAT), glutathione peroxidase (GSH-Px) and total superoxide dismutase (T-SOD) and increased malondialdehyde compared to those in the house or polytunnel. In relation to immune responses in the lambs’ serum, in the polytunnel, immunoglobulin A (IgA), tumor necrosis factor-α (TNF-α) and interleukin-4 (IL-4) were higher and immunoglobulin G (IgG) lower compared with the concentrations in lambs that were outdoors. Over the course of the experiment, genes expressing heat shock proteins and antioxidant enzymes increased in lambs in the outdoor treatment, whereas they decreased in lambs in the indoor treatments. It is concluded that although there were no treatment effects on core body temperature, the trends for progressive changes in blood composition and gene expression indicate that the outdoor lambs were not physiologically stable; hence, they should not be kept outdoors in these environmental conditions for long periods.

## 1. Introduction

Inner Mongolia is the main area producing sheep meat in China, and it has severe cold, heavy snow and strong winds for nearly half the year. Cold stress induced by low temperatures in winter is a very common challenge for sheep in feedlots. It reduces growth and feed efficiency and increases energy consumption, sometimes leading to death if the metabolic heat production is insufficient to maintain body temperature [1,2,3,4,5]. Acute cold exposure improves a sheep’s adaptive capacity, but long-term chronic cold exposure disturbs metabolism and thermal homeostasis, increases the production of reactive oxygen species (ROS), causes oxidation imbalance [6,7], reduces immune cytokine levels and immunoglobulin secretion and suppresses immune function [8,9,10,11]. Heat shock protein (HSP) is a molecular chaperone protein that plays an important role in defending the animal from inflammation and oxidative imbalance by maintaining cytoarchitecture and signal transduction [12,13].

In winter, providing sheds for sheep to withstand the harsh environment, especially when low temperatures are coupled with heavy snowfall and cold winds, can alleviate cold stress and improve animal welfare [14]. Keeping ewes in warm sheds during winter and spring can improve the lambing rate and the lambs’ growth rates [15]. Compared to winter grazing, providing conserved feed indoors enhances their welfare, as evidenced by faster growth and better physiological and biochemical characteristics in the blood [16]. Oat hay is a suitable supplement to grazing in such conditions, as it can reverse a negative nitrogen balance and improve a sheep’s energy status [17]. Cold exposure not only reduces a sheep’s growth, but it also increases immune globulin levels to increase their ability to resist acute cold stress [18]. However, the consumption of large amounts of nutrients to counteract the cold may result in oxidative damage. In one study, antioxidant enzymes levels in sheep were increased in the cold, but the concentration of MDA, which reflects cell damage, was decreased [19]. However, there was no other evidence that the sheep experienced oxidative damage [19].

In Inner Mongolia, farmers with feedlots for fattening lambs mostly use enclosed houses. However, polytunnels with a transparent roof to allow maximum solar radiation to enter and walls to protect from wind and snow may provide a lower cost alternative that copes better with the extreme conditions, as it allows radiant heat from the sun to penetrate to the sheep. In the Inner Mongolian sheep production systems, lambs are usually weaned at two months of age, after which they may be particularly prone to cold stress. Therefore, we selected weaned two-month-old female lambs as the research subjects for a comparison of growth performance, immune function and antioxidant status in winter in enclosed houses, a polytunnel and outdoors.

## 2. Materials and Methods

### 2.1. Animals and Experiment Design

The experiment was conducted at the experimental farm of the Inner Mongolia Agricultural University, Hohhot, China. Sheep were cared for in accordance with the guidelines for animal experiments of the Inner Mongolia Agricultural University. The experimental protocol (No: NND20212037) was approved by the Institutional Ethics Committee of the Inner Mongolia Agricultural University. A total of 45 two-month-old Dorper × Mongolian crossbred female lambs (body weight 16.42 ± 2.52 kg) were randomly allocated to three treatments with five replications each and three lambs in each replicate. The three treatments were enclosed housing (House), outdoor pens (Outdoor) and a polytunnel (Polytunnel). The adjacent enclosed houses each had four breeze block walls, a door and a window on the front wall and an opaque roof. The polytunnel lambs were in individual pens in a metal-framed construction with translucent plastic sides and roof. Outdoor lambs were kept in pens in a relatively exposed open area with each pen delineated by a metal mesh fence.

All experimental lambs received the same total mixed ration offered twice daily (at 08:00 and 15:00 h) with residual feed weighed and recorded daily. The experimental diet of the lambs was formulated according to the nutritional requirements and physiological state of the sheep, as prescribed in American standards [20] (NRC 2007). The physical composition and nutrient levels are shown in Table 1. The lambs were given free access to water throughout the experiment. Lambs were weighed on d 1 and 28 of the experiment and recorded as initial body weight (IBW) and final body weight (FBW), respectively. Average daily gain (ADG) and average daily feed intake (ADFI) were determined from these measurements.

### 2.2. Measurement of Ambient Indexes

The environmental indicators, namely air temperature, relative humidity and wind speed, in the house, outdoor pens and polytunnel were measured at 06:00, 14:00 and 19:00 h daily by a digital thermometer (Digital Thermometer ar847+, Smart Sensor, Dongguan City, China) and a small vane anemometer (testo 416-Small vane anemometer, Testo SE & Co. KGaA, West Chester, PA, USA).

### 2.3. Blood Collection and Analysis

Two blood samples were collected into 10 mL non-heparinised tubes and 10 mL heparinized tubes from the jugular vein of each lamb at 08:00 h on day 7, 14, 21 and 28. The blood samples in the 10 mL tubes were centrifuged for 20 min at 3000× *g* to collect the serum and frozen at −20 °C to analyze the concentrations of INS, HSP70, HSP90, IL-1, IL-4, TNF-α, IgA, IgG, CAT, GSH-Px, T-SOD, T-AOC and MDA. The blood samples in the 10 mL heparinized tubes were centrifuged for 20 min at 3000× *g* to collect the leukocytes and frozen at −80 °C to determine the gene expression of HSF-1, HSP70, HSP90, Nrf2, CAT, GPX-Px, SOD_1_, SOD_2_, *NF-κBp50*, *NF-κBp65*, IL-1β, IL-4 and TNF-α. The levels of INS, HSP70, HSP90, IL-1, IL-4, TNF-α, IgA, IgG and IgM in serum were determined by lamb-specific ELISA kits (Ruixin Biological Technology Co., Ltd., Quanzhou, China) according to the manufacturer’s instructions. The concentrations of CAT, GSH-Px, T-SOD, T-AOC and MDA were determined with commercial kits (Nanjing Jiancheng Institute of Bioengineering, Nanjing, China).

### 2.4. Total RNA Extraction and Quantitative RT-PCR Analysis

Total RNA in the leukocytes was extracted using TRIzol reagent (Invitrogen, Carlsbad, CA, USA) and then the isolated RNA was quantitatively and qualitatively determined as described previously [21]. RNA was reverse transcribed into cDNA using the PrimeScript RT Reagent Kit after removing genomic DNA contamination (Takara Bio Inc., Otsu, Japan), according to the manufacturer’s instructions. Quantification of the cDNA transcript was performed using a qPCR TB Green Kit with the gene-specific primer on the LightCycler 96 real-time PCR system. qRT-PCR was performed using 20 μL reactions and performed with cycling conditions as described previously [21]. The optimum annealing temperatures for the different genes are given in Table 2, and they were designed and synthesized by Shanghai Sangon Biotech (Shanghai, China). β-actin was used as a reference gene and the relative transcript quantities in the mRNA expression level were calculated using the 2^−ΔΔCT^ method.

### 2.5. Statistical Analysis

Data were analyzed by one-way ANOVA using the GLM procedure in SAS (Version 9.2; SAS Institute, Inc., Cary, NC, USA). The multiple comparisons between treatments were conducted by Duncan’s multiple range test (DMRT) and considered significant at *p* < 0.05. Data are presented as mean ± SD.

## 3. Results

### 3.1. Shelter Conditions

The environmental indicators, temperature, relative humidity, wind speed and NH_3_ and CO_2_ contents are shown for the three treatments in Table 3. The temperatures in the House and Polytunnel treatments were similar, and both were significantly higher than those in the Outdoor treatment. The relative humidities (RH) of the three treatments were significantly different, in the order RH_House_ > RH_Polytunnel_ > RH_Outdoor_. The wind speed in the outdoor treatment was significantly higher than those in the House or Polytunnel treatments. The level of CO_2_ in the Outdoor treatment was significantly lower than those in the House or Polytunnel treatments. NH_3_ concentrations of the three treatments were significantly different, in the order NH_3(House)_ > NH_3(Polytunnel)_ > NH_3(Outdoor)_.

### 3.2. Growth Performance and Body Temperature

The IBW, FBW and ADFI of the lambs were not statistically different between the treatments. However, the ADG of lambs in the House and Polytunnel treatments were similar and significantly greater than that of lambs in the Outdoor treatment (Table 4).

### 3.3. Body Temperature

Scalp and ear skin temperatures of lambs in the House and Polytunnel treatments were significantly greater than that of lambs in the Outdoor treatment. The rectal temperature of the lambs was not statistically different between the three treatments (Table 5).

### 3.4. Serum Insulin and Heat Shock Proteins

As shown in Table 6, the levels of insulin, HSP90 and HSP70 in the serum of the lambs in the House, Outdoor and Polytunnel treatments were unaffected by treatment on d 7 and d 14. On d 21, the serum HSP90 level of lambs in the polytunnel was significantly higher than that of lambs in the Outdoor treatment (*p* < 0.05), but neither were different from that of the lambs in the house. On day 28, compared to lambs in Outdoor treatments, the insulin level was significantly lower and HSP90 was significantly higher in the lambs in the polytunnel or house.

### 3.5. Serum Antioxidant Status Indicators

As shown in Table 7, serum CAT, GSH-Px, T-SOD, T-AOC and MDA levels of the lambs were not significantly different between the three groups on day 7. On d 14, serum T-SOD activity of the lambs in the Outdoor treatment was significantly lower than those in the other two groups (*p* < 0.05), and serum MDA content of the lambs was significantly higher than that in the polytunnel (*p* < 0.05), but not different from that in the house. On d 21 and 28, outdoor lambs had significantly lower levels of serum CAT, GSH-Px and T-SOD and significantly higher levels of MDA than those of lambs in the other two treatments. On d 28, outdoor lambs also had significantly lower levels of serum T-AOC than those in the other two treatments.

### 3.6. Immunity indicators

Serum interleukins and immunoglobulins of the lambs in the House, Outdoor and Polytunnel treatments are shown in Table 8. On d 7 and 14, serum IgA, IgG, IgM, IL-1β, IL-4 and TNF-α levels of the lambs were not significantly different between the three treatments (*p* > 0.10). On d 21 and 28, serum IgG level was higher for lambs in the Polytunnel than those in the Outdoor and House treatments, and also on d 21, the level of TNF-α was lower for lambs in the Outdoor treatment than lambs in the Polytunnel or House treatments. On d 28, IgA and IgG levels were higher for lambs in the Polytunnel, compared with lambs in the House and Outdoor treatments, which were similar. Also, on d 28, IL-4 level was higher in lambs in the Polytunnel treatment, lowest in lambs in the Outdoor treatment and intermediate in lambs in the House treatment, and TNF-α level was higher for lambs in the House and Polytunnel treatments compared with lambs in the Outdoor treatment.

### 3.7. The Relative Expression of mRNA

As shown in Figure 1, Figure 2 and Figure 3, over the course of the experiment, lambs in the Outdoor treatment gradually increased the gene expression of *HSF-1* and *Nrf2* compared with the other two groups (*p* < 0.05). From d 14, lambs in the Outdoor treatment also increased the gene expression of *HSP90*, *HSP70*, *Nrf2* and *SOD1* (*p* < 0.05) and decreased the gene expression of *RIP2*, *NF-κBp50*, *NF-κBp65* and *IL-1β* compared with the other two treatments; they also increased the gene expression of *CAT*, *GSH-Px* and *SOD2* compared with the Polytunnel (*p* < 0.05). On days 21 and 28, the Outdoor treatment increased the gene expression of *HSF-1*, *HSP90*, *HSP70*, *Nrf2*, *CAT*, *GSH-Px*, *SOD1* and *SOD2* and decreased the gene expression of *RIP2*, *NF-κBp50*, *NF-κBp65*, *IL-1β*, *IL-4* and *TNF-α,* compared with the other two treatments. Collectively, with the increase of cold exposure over time, *HSP* and antioxidant enzyme-related gene expression showed an increasing trend, and cytokine-related gene expression showed a decreasing trend.

## 4. Discussion

Continuous cold exposure in winter is an important cause of reduced animal welfare; and low profitability of the farming system. The different accommodation systems on farms and feedlots may affect the growth and health of sheep. Therefore, the House, Outdoor and Polytunnel treatments were designed to evaluate their effects on the growth performance, immune function and antioxidant status of young female lambs in winter in northern China.

### 4.1. Environmental Parameters, Growth and Temperature

Keeping lambs outdoors reduced their weight gain and adversely impacted on immunocompetence and antioxidant status, while the indoor systems (House and Polytunnel) alleviated the above situation. The polytunnel probably had greater effects on the lambs’ immune function than the house or outdoors because it provided exposure to radiant heat. Over the course of the experiment, the fact that the gene expressions of HSP and antioxidant enzymes increased, and the gene expressions of inflammatory cytokines decreased, suggests that there were cumulative effects of cold exposure.

Average gaily gain (ADG) and the scalp and ear skin temperature of lambs in the Outdoor treatment were lower than those in the House and Polytunnel. In the outdoors, the low temperature and increased wind speed environment would have increased the heat loss of the lambs, leading to reduced skin temperatures and a reduced growth rate, because energy consumption would have been increased to maintain body temperature. There was no difference between treatments in core body temperature, indicating that the sheep were relatively stable thermally; however, the cumulative biochemical effects on their blood and gene expression parameters indicate that had the experiment progressed beyond 28 d, there may have been more serious consequences.

The lower critical temperature of sheep is about −3 °C, depending on factors such as fleece length, wind speed, etc., and for every 1 °C drop from −3 °C, the daily maintenance metabolizable energy requirement of a 60 kg sheep increases by 0.14~0.64 MJ [22]. A previous study in a Tibetan winter found that the growth rate of ewes raised outdoors decreased by 28 g/d, compared with ewes raised in a shed [16]. Similarly, the ADG and feed efficiency of cattle has been reported to be lower in March, compared with a warmer April [2,3]. The air temperatures in our house and polytunnel were similar and about 10 °C higher than that of the outdoors, and the wind speed was highest outdoors. The resulting wind chill was sufficient to reduce average daily gain, but this was not compensated by increased feed intake. It is likely that the diet was not of sufficiently high nutrient concentration to allow the lambs to increase intake to counteract the cold without reducing growth rates. The polytunnel had relatively warm sunlight entering, which kept radiant heat in the house and reduced the energy loss caused by cold exposure, which was probably the main reason for the beneficial impact of the polytunnel on concentrations of immune and antioxidative indicators. NH_3_ and CO_2_ in livestock houses are also important factors affecting their health and growth. Enclosed sheep sheds that are poorly ventilated cause NH_3_ and CO_2_ accumulation in the shed, leading to a decline in immune and antioxidative function and an increase in body energy consumption for detoxification [23]. The lack of ventilation was evident, with increased concentrations of NH_3_ and CO_2_ in the house and polytunnel compared with those in the outdoor treatment. 

### 4.2. Gene Expression and Antioxidants

Heat shock proteins (HSPs) are a class of highly conserved molecular chaperone proteins that participate in signal transduction by regulating the structure and function of proteins [24]. Animals increase the level of HSP transcription to adapt to changes in the external environment [25]. Banerjee et al. [26] found that the gene expressions of *HSPA1A* and *HSPA8* in goat lymphocytes were increased significantly by low temperature in winter. Richi et al. [27] observed that the expression of the *HSP70* gene was higher in liver and kidney tissues, compared to heart and spleen, and the *HSP70* protein exhibited overexpression in the liver and heart tissues, compared to kidney and spleen tissues, in both the summer and winter seasons. In the present study, the levels of insulin, HSP70 and HSP90 in serum were not significantly different between the three treatments in the first two weeks. This may be because changes could not be programmed in the sheeps’ bodies within that time period. However, as the experiment progressed, the serum HSP90 content and *HSP90* mRNA expression in the leukocyte of the lambs in the Outdoor treatment were increased to maintain the intracellular structure and function to adapt to the cold environment [28]. The hypothalamus of livestock reacts to cold stress signals, activates the sympathetic nervous system, reduces serum insulin and increases serum glucose concentrations to compensate for increased energy expenditure [5]. The reduction in insulin releases *HSF-1* through insulin-like receptor decay accelerating factor 2 (*DAF-2*), and the nucleus and heat shock protein element (*HSE*) combine to increase the level of HSP transcription [29]. Therefore, compared with lambs in the house and polytunnel, the lower serum insulin levels and higher HSP levels appear to have been caused by greater cold stress in the lambs in the outdoor treatment. This result suggests that there were hormonal changes caused by the low temperatures which affected the expression of molecular chaperones.

Reactive oxygen species (ROS) are free radicals produced by the oxidative respiration of cell mitochondria. They are usually kept at a low level by the cellular antioxidant defense mechanism. However, the enhanced metabolism caused by the low temperatures would have produced a large amount of ROS and oxidized unsaturated fatty acids to increase MDA production. The latter was increased in the lambs kept outdoors by almost 50% by the end of the experiment. The antioxidant enzyme system can remove MDA to protect the body against oxidative stress. During cold exposure, serum GSH-Px and CAT activities of the lambs are decreased while MDA content increases [6]. Chronic cold stress reduced the activity of CAT, GSH-Px and SOD in chicken heart tissue [7]. In this study, the increased serum MDA content of outdoor lambs appears to have reduced the activity of serum antioxidant enzymes. *Nrf2* is a key factor in the transcription of antioxidant enzyme genes. When cold stress causes oxidative imbalance, excess ROS and HSP90 can both interact with Kelch-like epichlorohydrin-related protein 1 (*Keap1*), dissociate the *Keap1-Nrf2* complex and release *Nrf2* into the nucleus to activate its downstream antioxidant genes transcription [12]. Frigault et al. [30] found that compared with squirrels reared at room temperature, the mRNA transcription level of Nrf2 in the heart tissue of thirteen hibernating ground squirrels at 4~6 °C was significantly higher, and the mRNA expression of Keap1 was significantly reduced. On the 7th day of this study, the expression of the *Nrf2* gene in outdoor lambs was significantly higher than those of the other two groups. As the low temperature exposure period advanced, the transcription of *Nrf2* and downstream antioxidant enzyme genes in blood leukocytes were significantly increased, which may be due to a massive release of ROS in blood, and the *HSP90* in leukocytes increased the *HSP90-Keap1* interaction and activated the transcription of *Nrf2* and its downstream antioxidant enzyme genes in the lambs in the outdoor treatment. However, it is worth noting that although the expression of antioxidant enzyme-related genes was up-regulated, the serum antioxidant enzyme activity was still decreased. This may be because oxidative stress can increase the expression of antioxidant genes as a compensatory protective effect [31]. The antioxidant enzymes catalyze increased production of H_2_O_2_, which leads to the inhibition of the activity of antioxidant enzymes. Similar results have been found in another study, in which the expression of *Nrf2* mRNA and protein increased in rat liver cells treated with garlic extract, but the activity of antioxidant enzymes was still lower than normal during oxidative stress [32]. Blood plays a key role in transporting nutrients in the organism: when outdoor lambs were subjected to oxidative stress, excessive MDA was produced in the organism and released into the blood, and antioxidant enzymes were transported to key organs, such as the heart and liver, leading to the decrease in the antioxidant enzyme activity in blood [33]. Therefore, the decrease in the level of antioxidant enzymes in serum in outdoor lambs indicated that the organism was in oxidative imbalance, and the overexpression of *HSP90* and *Nrf2* mRNA was the body’s protective mechanism against ROS damage. This study showed that outdoor lambs had lower serum antioxidant enzyme activity and higher MDA content, but increasing the temperature in the house and polytunnel could alleviate the oxidative and antioxidant imbalance in the young female lambs, This was especially true in the Polytunnel treatment, which had sunshine entering. 

### 4.3. Immune Function

Serum immunoglobulins secreted by plasma B cells play a key role in the defense against pathogens. Low temperature conditions usually cause a reduction of immunoglobulins secretion. Guo et al. [19] found that chronic cold stress significantly reduced the serum IgG content in sheep. In this study, as the experimental period progressed, polytunnel lambs had a higher IgG level than the other two groups, which might be increased by the high content of IL-4 [24]. Immune cytokines in blood can regulate the immune response, and their levels in serum can reflect the immune status. The pro-inflammatory cytokine TNF-α and the anti-inflammatory cytokine IL-4 are both cellular cytokines that are secreted by T lymphocyte subtypes Th1 and Th2 cells. Inflammatory action caused by cold stress led to an alteration in Th1- and Th2-type cytokines secretion in serum [34]. In our study, long-term cold exposure outdoors significantly reduced the level of cytokines in sheep serum [19]. Cold stress inhibits humoral immunity in rats and reduces the proliferation of IL-1β and IL-4 levels in serum [8]. We found that outdoor sheep serum had lower IL-4 content, indicating that the low temperature resulted in the dysregulation of the Th1/Th2 cytokines profiles, adversely affecting the Th1/Th2 balance and inhibiting cytokine secretion. The decrease in cytokine content might also be caused by the decrease in nuclear factor kappa-B (*NF-κB*) regulation of its gene expression. *NF-κB* is a key transcription factor related to immune responses. The nucleotide-binding leucine-rich repeat-containing (*NLR*) family member *NOD1* can combine receptor-interacting proteins 2 (*RIP2*), which activates the *NF-κB* pathway to increase the transcription of immune genes [13]. *HSP90* can maintain the stability of *NOD1* during low temperature, keeping it in an inhibited state and reducing the activation of *NF-κB* [35]. During cold stress, the mRNA expression of *NF-κB* and TNF-α in mouse hypothalamic cells is significantly down-regulated [36]. In this study, the expression of *HSP90* mRNA was increased, and the expression of *NF-κB* mRNA was inhibited by *NOD1*, thereby reducing the expression of *IL-1β* and *TNF-α* mRNA. The overexpression of *HSP* could help the organism maintain homeostasis [24]. The upregulation of *HSP* not only increases the expression of antioxidant enzymes regulated by *Nrf2* transcription [12] but also reduces immune responses regulated by *NF-ĸB* transcription [36], and *Nrf2* and *NF-κB* also influence each other. Recent studies showed that *Nrf2* inhibited the activation of the *NF-κB* pathway by increasing the transcription level of antioxidant enzymes, thereby reducing the ROS-mediated activation of *NF-κB* and the damage to the body caused by inflammation [37]. Moreover, HSP might be passively or actively released from the necrotic or stressed cells, respectively, and extracellular HSP can promote regulatory T cells, participate in antigen presentation and stimulate immune response to protect the immune system from immunosuppression [28]. Therefore, the decrease in cytokines caused by low temperature may be the result of *HSP* gene regulation, and the polytunnel had sunlight with radiant heat, which was beneficial to the immune function of young female lambs. The lambs in long-term cold exposure experienced inflammatory damage and anti-oxidant imbalance. Cold exposure impacted the expression of *HSP*, *Nrf2* and *NF-κB*, which regulates immune responses and antioxidant function.

## 5. Conclusions

Outdoor keeping of young female lambs in winter at an ambient temperature of −14 °C and 55% relative humidity reduced their growth rate, serum immunocompetence and antioxidant status, compared with those in an enclosed house or a polytunnel. Although core body temperature was not adversely affected in the outdoor treatment, the serum biochemistry and gene expression parameters progressively changed in this treatment over the course of the 28-day experiment, indicating a lack of homeostasis at these temperatures. The polytunnel treatment was more conducive to good immune function and maintained the antioxidant balance best in the young female lambs. Therefore, polytunnels are recommended as the best option for the welfare of these sheep in these conditions, followed by enclosed housing, with the worst being keeping them outdoors.

## Figures and Tables

**Figure 1 animals-14-00518-f001:**
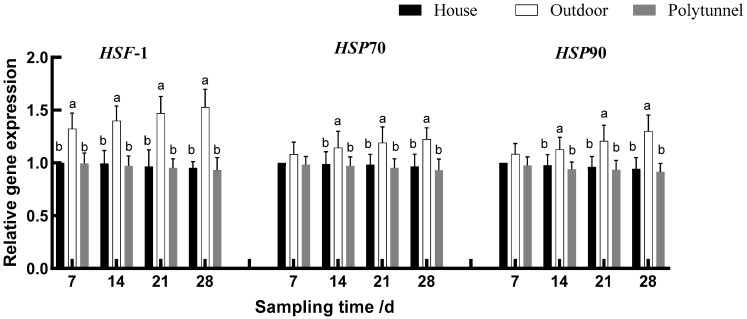
Effects of housing on *HSF*-*1*-, *HSP70*- and *HSP90*-related gene expression in young female lambs. Housed lambs’ gene expression on d 7 was taken as the calibrator. Data represent the means ± SD (n = 10). Means bearing different superscripts (a, b) differ significantly (*p* < 0.05) between treatments on the same day, *n* = 10.

**Figure 2 animals-14-00518-f002:**
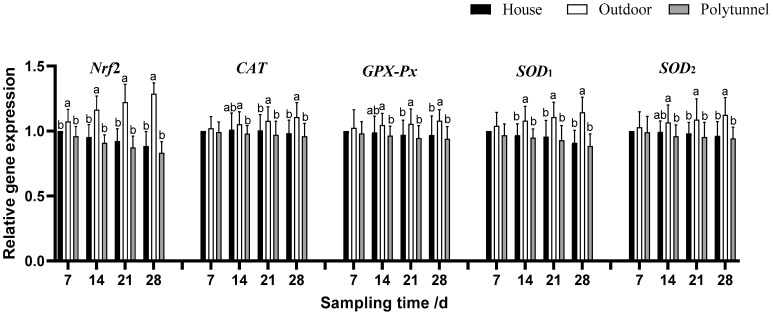
Effects of housing on *Nrf2-*, *CAT-*, *GPX-Px-*, *SOD_1_-* and *SOD_2_*-related gene expression in young female lambs. Housed lambs’ gene expression on d 7 was taken as the calibrator. Data represent the means ± SD (n = 10). Means bearing different superscripts (a, b) differ significantly (*p* < 0.05) between treatments on the same day, *n* = 10.

**Figure 3 animals-14-00518-f003:**
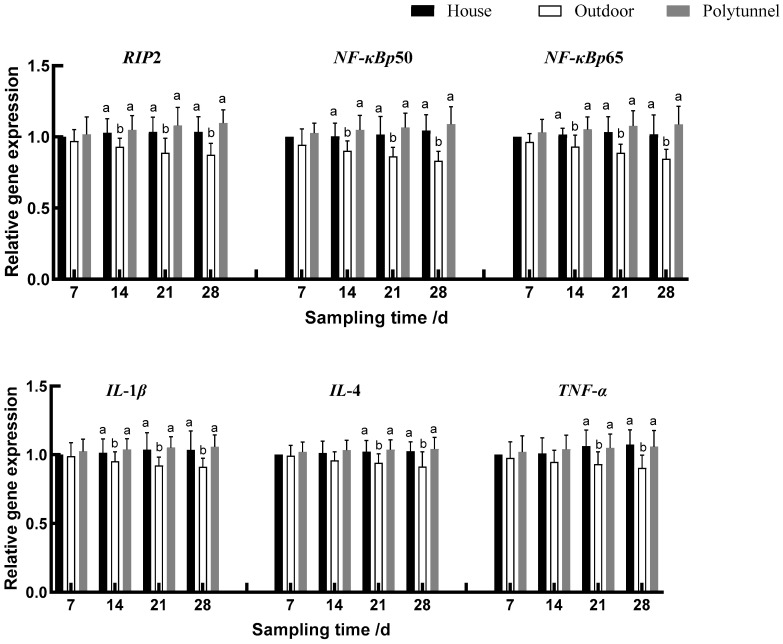
Effects of housing on *NF*-*κBp50*-, *NF*-*κBp65*-, *IL*-*1β*-, *IL-4-* and *TNF*-*α*-related gene expression in young female lambs. Housed lambs’ gene expression on d 7 was taken as the calibrator. Data represent the means ± SD. Means bearing different superscripts (a, b) differ significantly (*p* < 0.05) between treatments on the same day, *n* = 10.

**Table 1 animals-14-00518-t001:** Composition and nutrient level of the young female lambs’ diet (on an air-dried basis).

Physical Composition	%	Nutrient Composition	%, Unless Otherwise Stated
Corn	14.58	DE/(MJ/kg) ^2^	11.12
Asparagus	50.00	DM	90.72
Alfalfa hay	8.80	CP	13.62
Cottonseed meal	8.26	EE	1.96
Soybean meal	5.30	NDF	44.08
Wheat bran	8.36	ADF	29.08
DDGS	3.30	Ash	8.01
CaHPO_4_	0.12	Ca	0.83
NaCl	0.30	P	0.21
NaHCO_3_	0.48		
Premix ^1^	0.50		
Total	100		

DDGS: dried distillers’ grains with solubles; DE: digestible energy; DM: dry matter; CP: crude protein; EE: ether extract; NDF: neutral detergent fiber; ADF: acid detergent fiber. ^1^ The premix provided the following per kg of diet: vitamin A 6000 IU, vitamin D_3_ 2000 IU, vitamin E 15 IU, vitamin K_3_ 1.8 mg, vitamin B_1_ 0.35 mg, vitamin B_2_ 8.5 mg, vitamin B_6_ 0.9 mg, vitamin B_12_ 0.03 mg, D-pantothenic acid 16 mg, nicotinic acid 22 mg, folic acid 1.5 mg, biotin 0.15 mg, Cu 8 g, Fe 40 mg, Mn 20 mg, Zn 40 mg, I 0.8 mg, Se 0.3 mg, Co 0.3 mg. ^2^ DE was a calculated value, all others were measured values.

**Table 2 animals-14-00518-t002:** Genes and their primer sequence.

Gene	Gene Bank No.	Primer Sequence	Fragment Size	Annealing Temp (°C)
*β-Actin*	NM_001009784.1	F- AGCGCAAGTACTCCGTGTG	122	58
		R- CATTTGCGGTGGACGATG		
*HSP70*	JN604434.1	F- CCCACGAAGCAGACGCAGAT	66	52
		R-GCAGGTTGTTGTCCCGAGTCAT		
*HSP90*	EF091713.1	F-GCATTCTCAGTTCATTGGCTATCC	190	58
		R- GTCCTTCTTCTCTTCCTCCTCTTC		
*HSF-1*	XM_015097629.1	F-CAGCTGATGAAGGGGAAGCA	158	56
		R- TTTGACTGCACCAGCGAGAT		
*IL-1β*	NM_001009465	F- CGATGAGCTTCTGTGTGATG	161	59
		R-CTGTGAGAGGAGGTGGAGAG		
*IL-4*	AF1721681	F-GCTGAACATCCTCACATCGAG	87	60
		R- TTCTCAGTTGCGTTCTTTGG		
*TNF-α*	NM_001024860	F- AGTCTGGGCAGGTCTACTTTG	127	60
		R-GGTAACTGAGGTGGGAGAGG		
*NF-κBp50*	XM_004009667.3	F-AGCACCACTTATGACGGAACTACA	168	60
		R- GACCCCTTCATCCTCTCCATC		
*NF-κBp65*	XM_004020143.3	F-GGAGGCCAAGGAACTGAAGA	101	60
		R-TCAGGGGCAGAGGAAGGAG		
*RIP2*	NC_040260.1	F- CTCTGCGCTGTGTCCGTGTTC	219	60
		R-CAGGCTTCATCATCTGGCTCAC		
*Nrf2*	XM_004004557.1	F:TGTGGAGGAGTTCAACGAGC	88	61
		R:CGCCGCCATCTTGTTCTTG		
*CAT*	XM_004016396	F:GAGCCCACCTGCAAAGTTCT	148	60
		R:CTCCTACTGGATTACCGGCG		
*GSH-Px*	XM_004018462.1	F:TGGTCGTACTCGGCTTCCC	163	60
		R:AGCGGATGCGCCTTCTCG		
*SOD_1_*	NM_001145185	F:GGAGACCTGGGCAATGTGAA	182	60
		R:CCTCCAGCGTTTCCAGTCTT		
*SOD_2_*	NM_001280703.1	F:AAACCGTCAGCCTTACACC	116	60
		R:ACAAGCCACGCTCAGAAAC		

*β-actin*: beta-actin; *HSF-1*: heat shock transcription factor 1; *NF-κBp50*: nuclear factor kappa-B p50; *NF-κBp65*: nuclear factor kappa-B p65; *RIP2*: receptor-interacting proteins 2; *Nrf2*: nuclear factor erythroid 2-related factor 2; F: forward primer; R: reverse primer.

**Table 3 animals-14-00518-t003:** The environmental indicators of the three treatment groups during the test period.

Item	Treatment			*p*-Value
	House	Outdoor	Polytunnel	
Temperature (°C)	−4.90 ± 3.60 ^a^	−14.40 ± 4.70 ^b^	−3.65 ± 3.98 ^a^	0.001
Relative humidity (%)	69.13 ± 8.86 ^a^	55.21 ± 11.16 ^c^	62.28 ± 10.65 ^b^	0.001
Wind speed (m/s)	0.06 ± 0.02 ^b^	0.45 ± 0.43 ^a^	0.07 ± 0.03 ^b^	0.001
NH_3_ (mg/m^3^)	2.36 ± 0.83 ^a^	0.00 ± 0.00 ^c^	1.70 ± 0.99 ^b^	0.001
CO_2_ (mg/m^3^)	372.83 ± 73.24 ^a^	274.44 ± 21.43 ^b^	326.15 ± 98.90 ^a^	0.001

^a, b, c^ Different superscripts within the same row indicate a significant difference (*p* < 0.05). House: house; Outdoor: outdoor; Polytunnel: polytunnel.

**Table 4 animals-14-00518-t004:** Effects of housing on growth performance of young female lambs.

Items	Treatment			*p*-Value
	House	Outdoor	Polytunnel	
IBW (kg)	16.28 ± 2.66	16.61 ± 2.51	16.40 ± 2.56	0.901
FBW (kg)	18.61 ± 2.36	17.93 ± 2.61	18.99 ± 2.54	0.518
ADG (g/d)	83.17 ± 26.61 ^a^	47.19 ± 9.31 ^b^	92.50 ± 26.91 ^a^	0.001
ADFI (g/d)	577.40 ± 102.15	590.01 ± 123.05	633.68 ± 95.83	0.333

^a, b^ Different superscripts within the same row indicate a significant difference (*p* < 0.05). House: lambs in the house, Outdoor: outdoor lambs, Polytunnel: lambs in the polytunnel, *n* = 15. IBW: initial body weight, FBW: final body weight, ADG: average daily gain, ADFI: average daily feed intake.

**Table 5 animals-14-00518-t005:** Effects of housing on scalp and ear skin temperature and rectal temperature of young female lambs.

Items	Groups			*p*-Value
	House	Outdoor	Polytunnel	
Scalp temperature (°C)	13.66 ± 4.35 ^a^	10.39 ± 4.51 ^b^	12.83 ± 3.66 ^a^	0.001
Ear skin temperature (°C)	10.92 ± 4.11 ^a^	8.64 ± 4.69 ^b^	11.15 ± 4.69 ^a^	0.001
Rectal temperature (°C)	38.33 ± 0.54	38.37 ± 0.67	38.41 ± 0.63	0.770

^a, b^ Different superscripts within the same row indicate significant difference (*p* < 0.05), *n* = 10. House: house, Outdoor: outdoor, Polytunnel: polytunnel.

**Table 6 animals-14-00518-t006:** Effects of housing condition on serum insulin and heat shock protein level of young female lambs.

Items	Group			*p*-Value
	House	Outdoor	Polytunnel	
d 7				
INS (μIU/mL)	7.31 ± 0.41	7.84 ± 1.78	7.61 ± 2.02	0.623
HSP70 (pg/mL)	126.70 ± 24.91	134.60 ± 37.82	124.40 ± 18.21	0.648
HSP90 (pg/mL)	521.70 ± 52.88	564.60 ± 53.20	554.80 ± 53.31	0.219
d 14				
INS (μIU/mL)	7.55 ± 0.53	7.48 ± 1.94	7.91 ± 1.57	0.374
HSP70 (pg/mL)	127.60 ± 6.08	133.60 ± 18.26	125.40 ± 8.74	0.335
HSP90 (pg/mL)	523.10 ± 59.16	575.60 ± 63.38	536.80 ± 41.66	0.130
d 21				
INS (μIU/mL)	7.84 ± 0.96	7.12 ± 1.26	8.04 ± 1.49	0.080
HSP70 (pg/mL)	122.60 ± 16.15	129.50 ± 23.24	122.70 ± 10.45	0.617
HSP90 (pg/mL)	545.88 ± 71.24 ^ab^	593.96 ± 51.11 ^a^	531.78 ± 25.04 ^b^	0.046
d 28				
INS (μIU/mL)	7.87 ± 1.24 ^a^	6.75 ± 1.43 ^b^	8.31 ± 2.34 ^a^	0.012
HSP70 (pg/mL)	123.3 ± 8.80	128.0 ± 12.12	120.2 ± 7.24	0.259
HSP90 (pg/mL)	543.2 ± 38.42 ^b^	601.2 ± 63.41 ^a^	526.4 ± 49.87 ^b^	0.012

^a, b^ Different superscripts within the same row indicate a significant difference (*p* < 0.05), *n* = 10. House: house, Outdoor: outdoor, Polytunnel: polytunnel. INS: insulin, HSP70: heat shock protein 70, HSP90: heat shock protein 90.

**Table 7 animals-14-00518-t007:** Effects of housing on serum antioxidant concentrations in young female lambs.

Item	Group			*p*-Value
	House	Outdoor	Polytunnel	
d 7				
CAT (U/mL)	1.48 ± 0.14	1.41 ± 0.16	1.53 ± 0.17	0.400
GSH-Px (U/mL)	107.9 ± 15.22	102.8 ± 18.55	119.39 ± 17.88	0.266
T-SOD (U/mL)	126.50 ± 11.16	115.18 ± 15.66	128.88 ± 11.44	0.126
T-AOC (U/mL)	0.42 ± 0.08	0.37 ± 0.10	0.43 ± 0.06	0.365
MDA (nmol/mL)	2.17 ± 0.29	2.20 ± 0.44	2.10 ± 0.36	0.863
d 14				
CAT (U/mL)	1.46 ± 0.15	1.31 ± 0.20	1.56 ± 0.19	0.066
GSH-Px (U/mL)	115.04 ± 25.22	99.49 ± 20.86	127.29 ± 23.25	0.241
T-SOD (U/mL)	133.52 ± 11.13 ^a^	109.43 ± 15.30 ^b^	138.73 ± 16.78 ^a^	0.004
T-AOC (U/mL)	0.46 ± 0.04	0.42 ± 0.03	0.45 ± 0.04	0.118
MDA (nmol/mL)	2.10 ± 0.21 ^ab^	2.30 ± 0.25 ^a^	1.97 ± 0.44 ^b^	0.040
d 21				
CAT (U/mL)	1.43 ± 0.19 ^a^	1.22 ± 0.15 ^b^	1.58 ± 0.21 ^a^	0.015
GSH-Px (U/mL)	125.27 ± 11.85 ^a^	97.33 ± 21.86 ^b^	137.18 ± 14.65 ^a^	0.009
T-SOD (U/mL)	135.46 ± 14.84 ^a^	107.82 ± 14.12 ^b^	145.08 ± 18.39 ^a^	0.014
T-AOC (U/mL)	0.45 ± 0.07	0.40 ± 0.05	0.46 ± 0.04	0.054
MDA (nmol/mL)	1.94 ± 0.32 ^b^	2.52 ± 0.35 ^a^	1.88 ± 0.24 ^b^	0.004
d 28				
CAT (U/mL)	1.43 ± 0.22 ^a^	1.16 ± 0.17 ^b^	1.67 ± 0.20 ^a^	0.009
GSH-Px (U/mL)	132.17 ± 10.55 ^a^	93.25 ± 19.19 ^b^	148.15 ± 14.31 ^a^	0.001
T-SOD (U/mL)	136.84 ± 17.31 ^a^	104.02 ± 13.91 ^b^	151.07 ± 11.63 ^a^	0.006
T-AOC (U/mL)	0.44 ± 0.06 ^a^	0.39 ± 0.08 ^b^	0.45 ± 0.05 ^a^	0.063
MDA (nmol/mL)	1.91 ± 0.29 ^a^	2.70 ± 0.43 ^b^	1.82 ± 0.50 ^a^	0.001

^a, b^ Different superscripts within the same row indicate significant difference (*p* < 0.05), *n* = 10. House: lambs in the house, Outdoor: outdoor lambs, Polytunnel: lambs in the polytunnel. CAT: catalase, GSH-Px: glutathione peroxidase, T-SOD: total superoxide dismutase, T-AOC: total antioxidant capacity, MDA: malondialdehyde, d: day.

**Table 8 animals-14-00518-t008:** Effects of housing on serum interleukins and immunoglobulins in young female lambs.

Items	Group			*p*-Value
	House	Outdoor	Polytunnel	
7 d				
IgA (μg/mL)	47.06 ± 4.06	47.34 ± 4.74	48.05 ± 5.43	0.905
IgG (mg/mL)	12.35 ± 1.91	12.79 ± 2.93	12.43 ± 1.57	0.897
IgM (μg/mL)	352.91 ± 18.55	362.00 ± 31.08	362.62 ± 40.94	0.428
IL-1β (pg/mL)	12.37 ± 2.06	12.29 ± 2.21	11.82 ± 0.98	0.619
IL-4 (pg/mL)	10.55 ± 0.48	10.60 ± 1.00	10.12 ± 0.95	0.416
TNF-α (pg/mL)	26.70 ± 2.13	26.95 ± 2.64	27.22 ± 3.08	0.131
14 d				
IgA (μg/mL)	46.49 ± 2.37	46.80 ± 1.73	49.22 ± 2.76	0.054
IgG (mg/mL)	12.33 ± 1.08	12.67 ± 1.60	13.55 ± 0.96	0.108
IgM (μg/mL)	349.80 ± 18.50	352.02 ± 22.62	370.41 ± 20.59	0.296
IL-1β (pg/mL)	12.85 ± 0.68	11.80 ± 1.22	12.47 ± 0.95	0.093
IL-4 (pg/mL)	10.70 ± 1.32	10.13 ± 1.03	11.01 ± 1.28	0.294
TNF-α (pg/mL)	29.92 ± 0.78	26.61 ± 1.89	28.72 ± 3.68	0.158
21 d				
IgA (μg/mL)	43.64 ± 5.23	41.81 ± 2.29	44.82 ± 4.78	0.414
IgG (mg/mL)	12.77 ± 1.22 ^b^	12.05 ± 1.05 ^b^	13.67 ± 0.81 ^a^	0.030
IgM (μg/mL)	342.1 ± 25.41	336.4 ± 18.36	353.64 ± 45.29	0.588
IL-1β (pg/mL)	12.66 ± 1.06	12.74 ± 1.43	12.41 ± 1.13	0.826
IL-4 (pg/mL)	9.21 ± 1.16	8.33 ± 0.40	9.47 ± 1.15	0.069
TNF-α (pg/mL)	28.71 ± 2.58 ^a^	25.93 ± 1.84 ^b^	30.05 ± 2.96 ^a^	0.021
28 d				
IgA (μg/mL)	44.23 ± 3.30 ^b^	43.34 ± 2.58 ^b^	48.62 ± 2.62 ^a^	0.002
IgG (mg/mL)	12.85 ± 0.84 ^b^	12.56 ± 0.75 ^b^	14.66 ± 1.88 ^a^	0.006
IgM (μg/mL)	329.06 ± 38.01	323.43 ± 33.84	340.98 ± 22.26	0.770
IL-1β (pg/mL)	12.37 ± 2.06	12.51 ± 0.66	13.64 ± 1.25	0.207
IL-4 (pg/mL)	9.05 ± 0.75 ^ab^	8.62 ± 0.36 ^b^	9.65 ± 0.81 ^a^	0.016
TNF-α (pg/mL)	28.70 ± 2.13 ^a^	26.76 ± 1.80 ^b^	31.73 ± 2.49 ^a^	0.001

^a, b^ Different superscripts within the same row indicate significant difference (*p* < 0.05), *n* = 10. House: house, Outdoor: outdoor, Polytunnel: polytunnel. IgA: immunoglobulin A; IgG: immunoglobulin G; IgM: immunoglobulin M; IL-1β: interleukin 1β; IL-4: interleukin 4; TNF-α: tumor necrosis factor α; d: day.

## Data Availability

Data are available from the corresponding author on request.

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
