# Peer review of "The Effects of Housing on Growth, Immune Function and Antioxidant Status of Young Female Lambs in Cold Conditions"

_animals, 2024, doi:10.3390/ani14030518_

Round 1

Reviewer 1 Report

Comments and Suggestions for Authors

Overall comments:

The present work titled: Effects of Housing Conditions on Growth Performance, Immune Function and Antioxidant Status of Lambs in Winter subjected to peer review is a global investigation on lamb husbandry in Northern of China (Inner Mongolia) where the constraint of climate conditions is evident (long period of low temperatures with snow and wind). In this region, lamb feedlots constitute the main source of income for local farmers. These farmers use different shed strategies to face the harsh conditions. In the present paper, the authors are investigating the effects of three different housing conditions.

The findings confirmed the obvious effect of the closed housing conditions compared to the traditional open fenced housing method. Therefore, the obtained results were fully expected.

As a whole, the manuscript is fairly presented and written, but still required reediting and rectifications should be included.

Major comments:

The title largely reflects the content of the present study. However, it should be specified that the study concerned young lambs.

The abstract reported all the aspects of the study, but also contained some defects. In fact, the study was not focused on feeding conditions but on housing conditions and, therefore, it is not necessary to include the feeding qualification in specifying the three groups. In addition, growth performance should be added to key words.

The introduction is, somehow, convenient. It is mostly dedicated to the presentation of the environment of the study region and its climate and main popular activity. The given arguments were not sufficient to justify the scientific importance of carrying out such study on young female lambs.

Material & Methods:

This part is correctly structured. Firstly, it conveniently exposed data about the animals used in the present study and the experimental protocol adopted. The question which can be raised is why, in these conditions, using very young lambs? Then, the authors have given details about the parameters retained for this study including those related to climate data, to growth performance and those explored in blood samples (immune and antioxidant indexes), as well as the analyses techniques used. In my point of view, some parameters are overage and may be used in another specified complement study (in particular the gene expression of specified protein).

The used statistical tests were sufficiently defined.

Results:

The presentation of the obtained results is fairly convenient. The included Tables and Figures are adequate and explicitly report the data obtained. However, it is more convenient to separate Table 4 in two tables one for the “Effects of housing condition on scalp and ear skin temperature, rectal temperature in lambs and the other for the effects on growth performance.

The data related to growth performance, which is the main objective of the present study, appear somehow unpersuasive and unexpected since no significant difference has been noted.

Globally, the obtained results are expected and scientifically valuable; but cannot be totally conclusive, since the presented data concerned very young growing animals, particularly the low immune protection.

Discussion:

It needs to be revised and restructured. The obtained data were compared to some related studies in the same species, especially concerning the effect of low temperature without evoking the specificity of the present study. In fact, the age of lambs used, in this study, was not considered, particularly the immune protection status. Also, the origin of specimens should be accurately specified.

The conclusions are, somehow, speculative and subjective. A particular interest should be given the scientific value and practical significance of the obtained data. These conclusions should practically reply to the fixed objectives. No recommendations were issued from the present study.

The literature references are abundant and mostly recent.

Minor comments:

These comments are quoted in the enclosed version of the manuscript.(see attached file)

Comments on the Quality of English Language

The manuscript should be reedited

Author Response

Dear editors and reviewer,

Thank you very much for your precious comments on our manuscript entitled  “Effects of Housing Conditions on Growth Performance, Immune Function and Antioxidant Status of Lambs in Winter”. All of your comments have greatly contributed to the optimization of this manuscript. We have accepted all your comments and made careful revisions according to the comments, which have been included in the revised manuscript.

The details of the response to each reviewer ’s comment are as following:

Reviewer 1 Comments and Suggestions for Authors

Overall comments:

The present work titled: Effects of Housing Conditions on Growth Performance, Immune Function and Antioxidant Status of Lambs in Winter subjected to peer review is a global investigation on lamb husbandry in Northern of China (Inner Mongolia) where the constraint of climate conditions is evident (long period of low temperatures with snow and wind). In this region, lamb feedlots constitute the main source of income for local farmers. These farmers use different shed strategies to face the harsh conditions. In the present paper, the authors are investigating the effects of three different housing conditions.

The findings confirmed the obvious effect of the closed housing conditions compared to the traditional open fenced housing method. Therefore, the obtained results were fully expected.

As a whole, the manuscript is fairly presented and written, but still required reediting and rectifications should be included.

Major comments:

Comment 1: The title largely reflects the content of the present study. However, it should be specified that the study concerned young lambs.

Response: we specified that the study concerned young lambs in the revised manuscript in line 64 and highlighted in green. Young is a rather unclear term, in our view, as in some parts of the world the lambs might be much younger at weaning.

Comment 2: The abstract reported all the aspects of the study, but also contained some defects. In fact, the study was not focused on feeding conditions but on housing conditions and, therefore, it is not necessary to include the feeding qualification in specifying the three groups. In addition, growth performance should be added to key words.

Response: we rewrote the abstract in the new version, so the above problems are not in the revised abstract. But thanks for your recommendations.

Comment 3: The introduction is, somehow, convenient. It is mostly dedicated to the presentation of the environment of the study region and its climate and main popular activity. The given arguments were not sufficient to justify the scientific importance of carrying out such study on young female lambs.

Response: We did not emphasize the gender of lambs, just tried to offset the differences caused by gender, so we insisted on using lambs of the same gender.

Comment 4: Material & Methods:

This part is correctly structured. Firstly, it conveniently exposed data about the animals used in the present study and the experimental protocol adopted. The question which can be raised is why, in these conditions, using very young lambs? Then, the authors have given details about the parameters retained for this study including those related to climate data, to growth performance and those explored in blood samples (immune and antioxidant indexes), as well as the analyses techniques used. In my point of view, some parameters are overage and may be used in another specified complement study (in particular the gene expression of specified protein). The used statistical tests were sufficiently defined.

Response: Lambs are usually weaned at two months of age in Inner Mongolia, and their lower sebum thickness in young lambs compared to adult sheep. Therefore, we selected two month old lambs as the research object. And I am sorry that I did not understand “In my point of view, some parameters are overage and may be used in another specified complement study (in particular the gene expression of specified protein).”

Sorry, we do not understand the penultimate sentence; what is ‘overage’ in this context?

Comment 5: Results:

The presentation of the obtained results is fairly convenient. The included Tables and Figures are adequate and explicitly report the data obtained. However, it is more convenient to separate Table 4 in two tables one for the “Effects of housing condition on scalp and ear skin temperature, rectal temperature in lambs and the other for the effects on growth performance.

The data related to growth performance, which is the main objective of the present study, appear somehow unpersuasive and unexpected since no significant difference has been noted.

Globally, the obtained results are expected and scientifically valuable; but cannot be totally conclusive, since the presented data concerned very young growing animals, particularly the low immune protection.

Response: According to your comment, we separated Table 4 in two tables one for the “Effects of housing condition on scalp and ear skin temperature, rectal temperature in lambs and the other for the effects on growth performance. We transported the lambs from a different farm before the feeding experiment. And we used two- month-old lambs as we mentioned, so the environmental change are likely to be the main reasons for the results in growth rate. We also very dispointed at this results. We are completely agree with your comments.

Comment 6: Discussion:

It needs to be revised and restructured. The obtained data were compared to some related studies in the same species, especially concerning the effect of low temperature without evoking the specificity of the present study. In fact, the age of lambs used, in this study, was not considered, particularly the immune protection status. Also, the origin of specimens should be accurately specified.

Response: We have restructured the Discussion, which now contains sections.

Comment 7: The conclusions are, somehow, speculative and subjective. A particular interest should be given the scientific value and practical significance of the obtained data. These conclusions should practically reply to the fixed objectives. No recommendations were issued from the present study.

Response: according to your helpful comment, we moved“The lambs under long-term cold exposure caused inflammatory damage and anti-oxidant imbalance. Cold exposure impacted the expression of HSP, Nrf2 and NF-κB, which regulate immune responses and antioxidant function.”to the end of discussion. And we have added a brief recommendation in the conclusion.

Comment 8: The literature references are abundant and mostly recent.

Response: Thank you.

 Comment 8:These comments are quoted in the enclosed version of the manuscript.(see attached file)

Response: Thank you very much. We revised the manuscript according to your comments and highlighted key revised points in green. Please find them in the revised manuscript. We also rerevised all English writing of the manuscript. We didn’t highlight all revised content, because it was too much and tracked changes was leading to regular crashes.

Reviewer 2 Report

Comments and Suggestions for Authors

The overall logic of this manuscript is clear, and this study suggest that low temperature could reduce lamb’s daily gain, immune function and antioxidant status. closed house feeding and sunshine house feeding can alleviate the above situation, and sunshine house feeding is more benefit to the immune function. But there is also some deficiency enumerated below:

1.         It is suggested to supplement the significance of this article.

2.         During the experiment, the “Housing Conditions” were mixed with “low temperature” and “sunshine house feeding”. If we talk about low temperature and sunshine, it's 2 ×2 experiments, there should be four groups.

3.         In result 3.2, growth performance should be the main factor, while temperature related results should be evident.

4.         Tables 5, 6, and 7 have different styles.

5.         The discussion section should be simplified.

Comments on the Quality of English Language

 English language need be improved

Author Response

Dear editors and reviewer,

Thank you very much for your precious comments on our manuscript entitled  “Effects of Housing Conditions on Growth Performance, Immune Function and Antioxidant Status of Lambs in Winter”. All of your comments have greatly contributed to the optimization of this manuscript. We have accepted all your comments and made careful revisions according to the comments, which have been included in the revised manuscript.

The details of the response to each reviewer ’s comment are as following:

Reviewer 2 Comments and Suggestions for Authors

The overall logic of this manuscript is clear, and this study suggest that low temperature could reduce lamb’s daily gain, immune function and antioxidant status. closed house feeding and sunshine house feeding can alleviate the above situation, and sunshine house feeding is more benefit to the immune function. But there is also some deficiency enumerated below:

Comment 1: It is suggested to supplement the significance of this article.

Response: A brief the significance of this article “This study provides a theoretical basis for promoting the application of sunhouses in sheep breeding in cold regions” was added in line 82 in the revised manuscript. 

Comment 2: During the experiment, the “Housing Conditions” were mixed with “low temperature” and “sunshine house feeding”. If we talk about low temperature and sunshine, it's 2 ×2 experiments, there should be four groups.

Response: It is a very good idea, but “Housing Conditions” was the only one factor in this study, so we cannot revise the experiment design to a 2 factor factorial design.

Comment 3: In result 3.2, growth performance should be the main factor, while temperature related results should be evident.

Response: we separated Table 4 in two tables one for the “Effects of housing condition on scalp and ear skin temperature, rectal temperature in lambs and the other for the effects on growth performance. And the results discription also be seperated in two parts in line 174-182.

Comment 4: Tables 5, 6, and 7 have different styles.

Response: we revised the tables as requested in your comment, please find it in Table 6, 7 and 8 in the revised manuscript.

Comment 5: The discussion section should be simplified.

Response: we simplified the discussion as your comment, and created sections. Some content which was included in INTRODUCTION is removed. But we also added other content in the discussion according to the other reviewers’ requests, so it is not significantly less than before.

Comment 6: English language need be improved.

Response: we rerevised all English writing of all the manuscript. We didn’t highlight all revised content, because it was too much and the computer was frequently crashing.

Reviewer 3 Report

Comments and Suggestions for Authors

Dear Authors,

I read your manuscript with interest as it describes a valuable study, and it has to be further polished to provide the best possible reader understanding. Please find below my suggestions, observations and questions.

 Simple summary

L 14 – insert ‘the’ before ‘lamb’s’

L 15 – closed with capital “C”

L 16 – use either ‘has more benefits for’ or ‘is more beneficial to’ instead of ‘is more benefit to ‘; delete “And”

L 17 – ‘plays’ instead of ‘play’

 Abstract

Verify if the wordcount does not exceed the Journal’s instructions. Just as you began this section, all abbreviations have to be spelt out at their first usage and further in the section, this does not happen. If your wordcount is too high, the abbreviations could be removed here, then spelt out at their first usage at their next usage, but abbreviations without their spelt-out meaning should not exist here. Another suggestion would be to group the parameters tested and present them that way instead of the lists provided, for a more enjoyable reader experience at this stage (abstract).

 L 21 – delete ‘and’ before ‘the lambs were’

L 22 – delete ‘the’ before ‘blood samples’

L 23 – insert ‘and the’ before ‘scalp’, ‘were’ instead of ‘was’ before ‘lower’, insert ‘in’ before ‘those’

L 26 – insert ‘was more’ before ‘reduced’

L 27 –  insert ‘more’ before ‘increased’, replace ‘content’ with ‘concentration’

L 28 – replace ‘that’ with ‘in those’

L 29 – insert ‘in’ before ‘those’

L30 – verify the correctness of parameter names here (also consider the general suggestion for the section)

L34 – insert ‘the lambs in’ before OFG, replace ‘the’ with ‘their’ before ‘daily gain’

L36 – please rephrase the ‘stable state of the organism’ as it is not specific enough for a clear reader understanding; delete “While”

 Introduction

L42-45 – the paragraph could be more concise, focusing on the most relevant information for the reader

L47 – ‘exposure’ instead of ‘expose’ here and throughout the paper

L 48 – ‘leading’ instead of ‘which leads’

L49 – thermal homeostasis suggested

L52 – increases, causes

L 53 – reduces, suppresses

L55 – insert ‘which’ and delete the comma before ‘plays’. This sentence is disconnected from the one before and the other after it.

L58 – delete the comma and insert ‘and’ before ‘improve’

L60 – improve the lambing rate, daily weight gain, compared to

L 69-72 – reformulate

L70 – houses, the lambs, delete comma before ‘and’

 Materials and Methods

The English of the paper needs grammatical, orthographical and stylistic corrections, including the previous sections for which I provided several suggestions (which are not enough). I stop this type of suggestion here and this does not mean that corrections are not needed, but the opposite – a native (or at least well-experienced) English speaker must edit and proofread the manuscript.

L 155 – delete “extremely significant at p<0.01” because in the tables and figures you only mention p<0.05

-          Data with capital “D” (new sentence)

Results

L 159-161 – What data are these?

L 163 – delete the content in brackets (the data are in table 3)

Table 3 – space needed before the bracket “Wind speed(m/s)”

Table 4 – leave a space between the parameter and measure unit. Same in table 5 “INS(μIU/mL)”

L 179 – provide in caption the meaning for abbreviations (IBW, FBW, ADG and ADFI), just as you did for the other tables

L 203 – space needed before Table 6 (idem Table 7)

Discussion

L 254-268 – delete (you wrote this in other sections of the paper)

L 294- 300 – reformulate and split in several sentences

L 307, 309 – replace “&” with “and”

L – studies not Studies

L 354-359 – reformulate and split in several sentences

Conclusions

L 406-412 – reformulate and split in several sentences

L 412 – delete “But”

Comments on the Quality of English Language

Extensive editing of English language required

Author Response

Dear editors and reviewer,

Thank you very much for your precious comments on our manuscript entitled  “Effects of Housing Conditions on Growth Performance, Immune Function and Antioxidant Status of Lambs in Winter”. All of your comments have greatly contributed to the optimization of this manuscript. We have accepted all your comments and made careful revisions according to the comments, which have been included in the revised manuscript.

The details of the response to each reviewer ’s comment are as following:

Reviewer 3 Comments and Suggestions for Authors

Dear Authors,

I read your manuscript with interest as it describes a valuable study, and it has to be further polished to provide the best possible reader understanding. Please find below my suggestions, observations and questions.

Comment 1:

 Simple summary

L 14 – insert ‘the’ before ‘lamb’s’

L 15 – closed with capital “C”

L 16 – use either ‘has more benefits for’ or ‘is more beneficial to’ instead of ‘is more benefit to ‘; delete “And”

L 17 – ‘plays’ instead of ‘play’

Response: Thank you so much for your careful review, your comments contributed to our revision considerably. We rewrote the Simple Summary, please find attached the revised version, which takes account of your comments.

Comment 2:

 L 21 – delete ‘and’ before ‘the lambs were’

L 22 – delete ‘the’ before ‘blood samples’

L 23 – insert ‘and the’ before ‘scalp’, ‘were’ instead of ‘was’ before ‘lower’, insert ‘in’ before ‘those’

L 26 – insert ‘was more’ before ‘reduced’

L 27 –  insert ‘more’ before ‘increased’, replace ‘content’ with ‘concentration’

L 28 – replace ‘that’ with ‘in those’

L 29 – insert ‘in’ before ‘those’

L30 – verify the correctness of parameter names here (also consider the general suggestion for the section)

L34 – insert ‘the lambs in’ before OFG, replace ‘the’ with ‘their’ before ‘daily gain’

L36 – please rephrase the ‘stable state of the organism’ as it is not specific enough for a clear reader understanding; delete “While”

 Introduction

L42-45 – the paragraph could be more concise, focusing on the most relevant information for the reader

L47 – ‘exposure’ instead of ‘expose’ here and throughout the paper

L 48 – ‘leading’ instead of ‘which leads’

L49 – thermal homeostasis suggested

L52 – increases, causes

L 53 – reduces, suppresses

L55 – insert ‘which’ and delete the comma before ‘plays’. This sentence is disconnected from the one before and the other after it.

L58 – delete the comma and insert ‘and’ before ‘improve’

L60 – improve the lambing rate, daily weight gain, compared to

L 69-72 – reformulate

L70 – houses, the lambs, delete comma before ‘and’

Response: we rewrote the abstract in the new version, so the above problems are not in the revised abstract. But thanks for your recommendations.

Comment 3: Materials and Methods: L 155 – delete “extremely significant at p<0.01” because in the tables and figures you only mention p<0.05

Response: we deleted “extremely significant at p<0.01” from Statistical Analysis, and considered significant at p<0.05 in the revised manuscript.

Comment 4: -Data with capital “D” (new sentence)

Response: capital “D” was used in the new sentence according to your comment in line 157.

Comment 5: Results:L 159-161 – What data are these?

Response: the result was revised and these data were deleted.

Comment 6: L 163 – delete the content in brackets (the data are in table 3)

Table 3 – space needed before the bracket “Wind speed(m/s)”

Table 4 – leave a space between the parameter and measure unit. Same in Table 5 “INS(μIU/mL)”

L 203 – space needed before Table 6 (idem Table 7)

Response: all the brackets were deleted in Table 3 and a space between the parameter and measure unit was added in Table 5, 6,7 and 8 in the revised manuscript.

Comment 7: L 179 – provide in caption the meaning for abbreviations (IBW, FBW, ADG and ADFI), just as you did for the other tables

Response: Abbreviations (IBW, FBW, ADG and ADFI) are provided in line 186.

Comment 8: Discussion:L 254-268 – delete (you wrote this in other sections of the paper)

Response: according to your comment, we deleted this part in the revised discussion.

Comment 9: L 307, 309 – replace “&” with “and”   .

L – studies not Studies

Response: we replaced “&” with “and” and highlighted this in line 327 and 330.

Comment 10: L 294- 300 – reformulate and split in several sentences

Response: we rewrote the part and highlighted it in yellow in line 311-319.

Comment 11: L 354-359 – reformulate and split in several sentences

Response: we rewrote the part and highlighted it in yellow in line 373-379.

Comment 12: Conclusions: L 406-412 – reformulate and split in several sentences

L 412 – delete “But”

Response: we rewrote the conclusion. and deleted “But”.

Comment 13:  Abstract: Verify if the wordcount does not exceed the Journal’s instructions. Just as you began this section, all abbreviations have to be spelt out at their first usage and further in the section, this does not happen. If your wordcount is too high, the abbreviations could be removed here, then spelt out at their first usage at their next usage, but abbreviations without their spelt-out meaning should not exist here. Another suggestion would be to group the parameters tested and present them that way instead of the lists provided, for a more enjoyable reader experience at this stage (abstract).

Response: we rewrote the abstract. All abbreviations and the full name of the indexes were written as wrote in the simple summary in the revised manuscript.

Comment 14: The English of the paper needs grammatical, orthographical and stylistic corrections, including the previous sections for which I provided several suggestions (which are not enough). I stop this type of suggestion here and this does not mean that corrections are not needed, but the opposite – a native (or at least well-experienced) English speaker must edit and proofread the manuscript. Extensive editing of English language required.

Response: we rerevised all English writing of all the manuscript. We didn’t highlight all revised content, because it was too much.

Reviewer 4 Report

Comments and Suggestions for Authors

This study shows the adverse effects of open fenced management in an extremely cold environment, with several measures of production and animal welfare impaired by the inadequate shelter.

The benefits of the sunshine house feeding system are less obvious, with very little difference in temperature compared with a closed house, but there were some differences in IgA and IgG, indicating a benefit to the immune system of the sunshine housing.

The report covers a large range of biological parameters measured, and the authors comprehensively discuss the effects of cold on these values, and the various interactions between them.

Comments on the Quality of English Language

The manuscript requires editing by a fluent English user. There are many cases of poor phrasing, particularly in the use of past tense when present tense is required, or vice versa. However, the intended meaning is usually clear.

Author Response

Dear editors and reviewer,

Thank you very much for your precious comments on our manuscript entitled  “Effects of Housing Conditions on Growth Performance, Immune Function and Antioxidant Status of Lambs in Winter”. All of your comments have greatly contributed to the optimization of this manuscript. We have accepted all your comments and made careful revisions according to the comments, which have been included in the revised manuscript.

The details of the response to each reviewer ’s comment are as following:

Review 4  Comments and Suggestions for Authors

Comment 1:This study shows the adverse effects of open fenced management in an extremely cold environment, with several measures of production and animal welfare impaired by the inadequate shelter. The benefits of the sunshine house feeding system are less obvious, with very little difference in temperature compared with a closed house, but there were some differences in IgA and IgG, indicating a benefit to the immune system of the sunshine housing.

Response: according to your comment, we tried to emphasize the benefits of the polytunnel in the revised manuscript, and highlighted the content in blue, such as in the abstract, results, discussion and conclusion.

Comment 2: The report covers a large range of biological parameters measured, and the authors comprehensively discuss the effects of cold on these values, and the various interactions between them.

Response: Thank you 

Comment 3: The manuscript requires editing by a fluent English user. There are many cases of poor phrasing, particularly in the use of past tense when present tense is required, or vice versa. However, the intended meaning is usually clear.

Response: we revised all English writing of all the manuscript. We didn’t highlight all revised content, because it was too much and led to frequent crashes of the computer.

Round 2

Reviewer 1 Report

Comments and Suggestions for Authors

I appreciate the effort made by the authors to improve the content of this work. However, I insist to include the qualification of young female lambs in the title , key words and the whole text. The revised document has improved the content of this study. 

Comments on the Quality of English Language

The quality of editing language is largely improved.

Author Response

Overall comments: I appreciate the effort made by the authors to improve the content of this work. However, I insist to include the qualification of young female lambs in the title , key words and the whole text. The revised document has improved the content of this study. 

Response: Thank you very much for your approval of our work. According to your comment, we added “young female” lambs in the title , key words and the whole text and highlighted it in blue to makes it easier for you to find.

Reviewer 3 Report

Comments and Suggestions for Authors

Dear Authors,

I am glad for all the improvements in your corrected paper. Please find my suggestions below.

 Simple summary

I am still not pleased by the use of abbreviations without spelling them at first occurrence… Cannot be a collective term used for the parameters somehow?

Something wrong happened with your manuscript as the Abstract is partly cut and only parts of sentences show, please correct this.

 General comment: please be more concise and avoid repetitions! Please take care to spell out the abbreviations at their first usage everywhere, throughout the paper.

L426: indicate (not indicates)

The conclusions are not the best possible. The newly introduced two sentences are correct English, but the style can be improved. The first part of the conclusion is repetitive, could you rephrase it? I think of switching from a restrictive standpoint (should not be kept outdoors… etc) to stating what is the better practice. Emphasize what would be better in the light of your results. To provide pro-welfare conditions, the lambs should be kept inside (house, polytunnel) in weather conditions which are similar to the studied ones – but be more specific about the conditions, state specifically the cold weather (‘temperatures less than…’ or similar), then give an allowance. If there is no other possibility, the outdoor housing at these temperatures should not exceed a maximum period of 28 days. Your research has shown that even if 28 days seems clinically safe, the response of the lambs’ organism is clear and provable.

I hope that my comments are helpful.

Comments on the Quality of English Language

Minor editing of English language required

Author Response

Overall comments: Dear Authors,

I am glad for all the improvements in your corrected paper. Please find my suggestions below.

 Simple summary

I am still not pleased by the use of abbreviations without spelling them at first occurrence… Cannot be a collective term used for the parameters somehow?

Response: we cannot use a collective term as some parameters were increased and some decreased. We have added all abbreviations in full at first occurrence in the abstract and highlighted them in yellow.

Something wrong happened with your manuscript as the Abstract is partly cut and only parts of sentences show, please correct this.

Response: this has been corrected.

 General comment: please be more concise and avoid repetitions! Please take care to spell out the abbreviations at their first usage everywhere, throughout the paper.

Response: we have further revised the ms to avoid repetition.

L426: indicate (not indicates)

Response: this has been checked and revised.

The conclusions are not the best possible. The newly introduced two sentences are correct English, but the style can be improved. The first part of the conclusion is repetitive, could you rephrase it? I think of switching from a restrictive standpoint (should not be kept outdoors… etc) to stating what is the better practice. Emphasize what would be better in the light of your results. To provide pro-welfare conditions, the lambs should be kept inside (house, polytunnel) in weather conditions which are similar to the studied ones – but be more specific about the conditions, state specifically the cold weather (‘temperatures less than…’ or similar), then give an allowance. If there is no other possibility, the outdoor housing at these temperatures should not exceed a maximum period of 28 days. Your research has shown that even if 28 days seems clinically safe, the response of the lambs’ organism is clear and provable.

Response: Based on your comments, we updated the conclusions extensively and highlighted this part in yellow, please find it in the manuscript R2.

I hope that my comments are helpful.

Response: thank you; they have improved the ms.

Minor editing of English language required

Response: English language has been edited by Prof. Clive J. C. Phillips, who is a native speaker at Curtin University, Australia.